# The Role of the Immune Checkpoint Molecules PD-1/PD-L1 and TIM-3/Gal-9 in the Pathogenesis of Preeclampsia—A Narrative Review

**DOI:** 10.3390/medicina58020157

**Published:** 2022-01-20

**Authors:** Johanna Mittelberger, Marina Seefried, Manuela Franitza, Fabian Garrido, Nina Ditsch, Udo Jeschke, Christian Dannecker

**Affiliations:** Department of Obstetrics and Gynecology, University Hospital Augsburg, Stenglinstraße 2, 86156 Augsburg, Germany; johanna.mittelberger@uk-augsburg.de (J.M.); marina.seefried@uk-augsburg.de (M.S.); manuela.franitza@uk-augsburg.de (M.F.); fabian.garrido@uk-augsburg.de (F.G.); nina.ditsch@uk-augsburg.de (N.D.); christian.dannecker@uk-augsburg.de (C.D.)

**Keywords:** preeclampsia, immune checkpoint molecules, PD-1, PD-L1, TIM-3, Gal-9

## Abstract

Preeclampsia is a pregnancy-specific disease which is characterized by abnormal placentation, endothelial dysfunction, and systemic inflammation. Several studies have shown that the maternal immune system, which is crucial for maintaining the pregnancy by ensuring maternal-fetal-tolerance, is disrupted in preeclamptic patients. Besides different immune cells, immune checkpoint molecules such as the programmed cell death protein 1/programmed death-ligand 1 (PD-1/PD-L1 system) and the T-cell immunoglobulin and mucin domain-containing protein 3/Galectin-9 (TIM-3/Gal-9 system) are key players in upholding the balance between pro-inflammatory and anti-inflammatory signals. Therefore, a clear understanding about the role of these immune checkpoint molecules in preeclampsia is essential. This review discusses the role of these two immune checkpoint systems in pregnancy and their alterations in preeclampsia.

## 1. Introduction

Preeclampsia (PE) is one of the most challenging and dangerous complications in pregnancy. It is defined by new onset of hypertension in pregnancy combined with proteinuria or another significant organ disfunction after 20 weeks of gestation. When occurring it can lead to serious complications such as renal failure, liver damage, HELLP-syndrome (hemolysis, elevated liver enzyme levels, low platelet count) or disfunction in the central nerval system, which all can result in high maternal and fetal mortality. In addition, the disease can result in a higher risk of cardiovascular or renal disease in women suffering from preeclampsia. The only effective treatment is the delivery of fetus and placenta which needs to be thoroughly evaluated considering the possible maternal and fetal complications of early delivery. Up until now the exact pathogenesis of this disease is still under debate and not fully understood. The placenta with its abnormal placentation and elevated levels of antiangiogenic factors seems to play a central role in the emergence, which then leads to a systemic inflammation and development of endothelial dysfunction [1,2].

During pregnancy, well matched and adjusted immune responses are essential for formation and maintenance of the pregnancy without rejecting the fetus [3]. Studies have shown that deviant reactions of the immune system play an important role in the pathogenesis of preeclampsia. Because immune checkpoint molecules such as the programmed cell death protein 1/programmed death-ligand 1 (PD-1/PD-L1) system and the T-cell immunoglobulin and mucin domain-containing protein 3/Galectin-9 (TIM-3/Gal-9) system are decisive in the regulation of immune responses, we aim to highlight their role in the pathogenesis of preeclampsia in this review.

## 2. Materials and Methods

PubMed was searched for articles concerning the immune checkpoint molecules PD-1/PD-L1 as well as TIM-3/Gal-9 in human placenta and in preeclampsia on 21 October 2021. A total of 59 publications resulted from that search. Eligible for inclusion were original articles including experimental studies, clinical trials, case-control studies, and cohort studies that examined the status of the PD-1/PD-L1 and the TIM-3/Gal-9 system in human placenta and their role in the pathogenesis of preeclampsia. This resulted in an inclusion of 30 publications which are summarized below.

## 3. The PD-1/PD-L1 System in Normal Pregnancy and Preeclampsia

### 3.1. The PD-1/PD-L1 System in Normal Pregnancy

The transmembrane receptor PD-1 as one of the two key-players in the PD-1/PD-L1 system is located on various cells of the immune system, among others on antigen presenting cells, T cells, B cells, and natural killer (NK) cells [4,5,6]. Its ligand PD-L1 can be found in different tissues like for example the placenta as well as on various immune cells like dendritic cells, macrophages, and T cells [4,7,8]. The binding of PD-L1 to its receptor PD-1 causes an inhibition of T-cell proliferation, down-regulation of pro-inflammatory T-cell activity and an inhibition of cytokine production and cytolytic function [9,10,11]. It stimulates the formation of regulatory T cells (Tregs), enhances their function, and also causes an inhibition of T effector cells. By these mechanisms a receptor activation leads to an inhibiting effect on the immune system [12].

Several studies can be found concerning the PD-1/PD-L1 system in murine pregnancy. The PD-1 expression is present on a variety of different decidual lymphocytes. Among others these contain NK-cells, NKT-like cells, T helper cells, and CD4+/CD8+ T cells. An increased PD-1 expression leads to a reduction in cytolytic activity of these cells. The ligand PD-L1 can be found on the decidua basalis as well as on the syncytiotrophoblast [13,14]. Blocking the ligand PD-L1 in murine pregnancy leads to an “increased fetal resorption rate and a reduction in the litter size” and thereby to decreased fetal survival rates [9,13,15]. Looking at the maternal-fetal-interface a PD-L1 blockade resulted in infiltration of T cells as well as higher levels of interferon (IFN)-γ and complement deposits. This suggests local T-cell-mediated rejection mechanisms. PD-L1-deficient pregnant mice also showed a reduction in regulatory T cells with an induction of apoptosis of Tregs as well as a shift toward a higher frequency of Th17 cells. An elevated number of Th17 cells after PD-L1 blockade leads to a higher level of Interleukin 17 and thereby results in activation of neutrophils and inflammation [9,15,16]. In comparison, another experiment of Taglauer et al. showed no difference in gestation length, litter size, or pup weight when PD-1 or PD-L1 is absent [17].

The data and information about the status of the PD-1/PD-L1 system during human pregnancy are poor. Enninga et al. showed elevated blood levels of soluble PD-L1 (sPD-L1) in pregnant women compared to healthy non-pregnant controls, which may result in suppressed maternal immunity [18,19]. The levels increased throughout gestation. Levels of Gal-9, another important immune checkpoint molecule, was also shown to be elevated in maternal blood during pregnancy. In comparison to sPD-L1 it did not increase throughout pregnancy but remained on an elevated level. The study also demonstrated that carrying a male fetus was accompanied with significantly higher levels of Gal-9, but not sPD-L1, compared to a female fetus [18].

A study performed by Meggyes et al. 2019 found elevated PD-1 expression in CD8+ T cell, CD4+ T cells, and NKT-like cells in the decidua of pregnant women. The PD-L1 levels in CD4+ T cells, Tregs, and NKT-like cells were also shown to be elevated. The cytotoxicity of PD-1 negative decidual immune cells was higher compared to peripheral blood, but the cytotoxicity in the decidual PD-1 positive CD8+ T cells was lower in comparison [20].

Meggyes et al. conducted another study analyzing the PD-1/PD-L1 status on CD4+ cells/CD8+ cells/Treg cells and NKT-like cells in peripheral blood. It showed a significantly higher expression of PD-1 on CD8+ cells in the second trimester compared to the first and third trimester while the PD-1 expression on the other cells did not differ compared to non-pregnant controls. The PD-L1 expression on CD4+ and CD8+ cells was significantly reduced in the third trimester compared to first and second trimester. Looking at the PD-L1 expression on Tregs the study showed a significantly higher expression in first and second trimester compared to non-pregnant controls [21].

### 3.2. The PD-1/PD-L1 System in the Human Placenta

When looking at the human placenta, Petroff et al. were the first to show high expression of PD-L1 protein in syncytiotrophoblast and extravillous cytotrophoblasts compared to a lack of the protein in placental stroma cells including macrophages. The levels of PD-L1 increased throughout the pregnancy [7].

To determine the status of PD-L1 expression in trophoblastic cells during normal pregnancy as well as in gestational trophoblastic diseases, Veras et al. used an anti-PD-L1-specific antibody to perform immunohistochemistry on placenta and tumor tissue. Concerning normal placentas, the study showed a high expression of PD-L1 in the syncytiotrophoblast. Lower PD-L1 expression was shown in the implantation site as well as in trophoblastic cells in the chorion leave. Looking at the cytotrophoblastic cells no PD-L1 expression was detectable. These mechanism of locally upregulating PD-L1 expression might contribute to the feto-maternal tolerance [8].

Similar results were shown by Lu et al. who performed immunohistochemistry on trophoblastic tissue and tumors as well. PD-L1 expression was strong in the syncytiotrophoblast and weak/negative in the cytotrophoblast. A moderate PD-L1 staining was shown in intermediate trophoblastic cells in normal placenta [22].

A study performed by Mincheva-Nilsson et al. investigated exosomes, which are small vesicles secreted from different cells to transport proteins and nucleic acid as well as communicate with neighboring cells. They demonstrated that these exosomes transport PD-L1 after being released from the placenta. Its suggested role is to downregulate immune effector cells by binding to its receptor PD-1 on maternal immune cells [23].

Zhang et al. demonstrated a promotion of sPD-L1 secretion by trophoblast cells after activation with IFN-β during pregnancy which then leads to a shift of macrophage polarization toward M2 phenotype, the anti-inflammatory state. This contributes to maternal-fetal tolerance by promoting a decrease in inflammation [24].

Using immunohistochemistry and PCR, van’t Hof et al. investigated the difference of PD-L1 and CD200 expression in the placenta between oocyte donation (OD) pregnancies and naturally conceived pregnancies with and without preeclampsia. The PD-L1 and CD200 expression was shown to be significantly decreased in OD pregnancies compared to normal pregnancies. Preeclamptic patients after oocyte donation showed lower PD-L1 and CD200 protein expression compared to naturally conceived patients. These alterations might contribute to the higher risk of developing preeclampsia after oocyte donation [25].

### 3.3. The PD-1/PD-L1 System in Preeclampsia

The influence of the PD-1/PD-L1 system in the pathogenesis of preeclampsia is the subject of current research. There are several studies which have shown an alteration of these immune checkpoint molecules in preeclamptic women.

Gu et al. investigated the status of soluble PD-1 and soluble PD-L1 in preeclampsia. The study showed significantly higher sPD-1 levels and relatively higher sPD-L1 levels in preeclamptic patients compared to normal pregnancies. The sPD-1 levels at <34 weeks of gestation were proven to be significantly higher in preeclamptic women compared to healthy pregnant women. Looking at pregnancies >34 weeks of gestation showed a relatively, but not significantly higher sPD-1 level in preeclampsia. To distinguish between fetal sex difference the groups were divided into female and male fetal sex. Looking at the sPD-1 and sPD-L1 levels revealed relatively higher levels of both proteins when carrying a female fetus. This study showed that the sPD-1/sPD-L1 system is aberrant in preeclampsia and thereby might contribute to an augmented immune response, although the exact role of these molecules remains unknown [26,27].

To investigate the possible role of the PD-1/PD-L1 system in the pathogenesis of early onset preeclampsia, Meggyes et al. performed a cross-sectional study on mononuclear cells in peripheral blood to compare their phenotype and functional characteristics. Comparing the PD-1 expression by different lymphocyte subjects between preeclamptic patients and healthy pregnancies in the third trimester revealed a significant upregulation of the receptor on CD8+ T cells, CD4+ T cells, Tregs and NKT-like cells in early onset preeclampsia. No difference was seen in the PD-1 expression by NK cells. Looking at the PD-L1 molecule on these cells showed significantly elevated levels on NKT-like cells in the preeclamptic group. The cytotoxicity of PD-1+ CD8+ T cells was significantly elevated in early onset preeclampsia, while the lytic activity of NKT-like cells was reduced regardless of their PD-1 status. The percentage of NKT-like cells co-expressing PD-1 and the NK cell activating receptor NKG2D was significantly higher in the preeclamptic study group while their cytotoxic activity was reduced. In comparison to Gu et al. [26] this work did not show increased levels of sPD-L1 in preeclampsia, which proves that further studies are needed. Concluding these results, the authors hypothesize that the PD-1 expression might be associated with the failure of the axis to diminish Th1 responses which might contribute to the elevated immunoactivation in preeclampsia [28].

In contrast, looking at the maternal-fetal interface, Morita et al. examined the T-cell receptors and the PD-1 expression on effector memory CD8+ T cells and naive CD8+ T cells in the peripheral blood and decidua in normal pregnancy and complicated pregnancy such as preeclampsia. Among other results, the study showed that PD-1 expression was evident on decidual CD8+ effector memory cells in normal pregnancy, whereas the PD-1 expression on these cells was downregulated in preeclampsia cases [29].

Tian et al. further studied the association between the Treg/Th17 imbalance, showed in preeclampsia, and the PD-1/PD-L1 system in peripheral blood samples of preeclamptic patients comparing them to normal pregnancies. The percentage of Treg cells was significantly decreased in preeclampsia, accompanied with a significantly higher expression of PD-1 on these cells. There was no difference in PD-L1 expression on Tregs. The number of Th17 cells was significantly increased in preeclamptic patients, leading to a lower Treg/Th17 ratio in this patient group compared to normal pregnancy. Looking at the PD-1-/PD-L1 status on Th17 cells showed a significantly lower expression of PD-1 and a significantly higher expression of PD-L1 in preeclampsia. Considering these results, the PD-1/PD-L1 expression might correlate with the Treg/Th17 imbalance seen in preeclampsia. The study group then did further research by using PE-like rat models which were also characterized by an imbalance of Treg/Th17. Treating them with the protein PD-L1-Fc lead to a reverse of the Treg/Th17 imbalance and thereby to a protective effect on mother and fetus. It also resulted in a protection of the placenta by undoing placental damages [30].

The study performed by Daraei et al. also investigated the status of PD-1 expression on peripheral Treg cells in preeclampsia. Similar to the previous studies they demonstrated a significantly reduced number of Treg cells and a significantly increased percentage of PD-1 expression on exhausted Tregs in preeclampsia. The authors concluded that, since PD-1 is involved in T-cell exhaustion, the presence of the molecule on Treg cells is associated with reduced function and thereby may contribute to the pathogenesis of preeclampsia [31].

Zhang et al. tried to find out more about the association between the Treg/Th17 imbalance and the PD-1/PD-L1 system as well. They discovered a decreased expression of PD-1, PD-L1, and Foxp3 (a nuclear transcription factor important for the quantity and function of Tregs and Th17 cells) in the placenta of preeclamptic patients compared to normal pregnancies. The expression of RORγt, another nuclear transcription factor, was increased in preeclampsia. They also showed that a decreased PD-1 expression might promote the proliferation of Th17 cells and inhibit the Treg cell differentiation and thereby contribute to a higher number of Th17 cells in preeclampsia. The administration of PD-L1 Fc led to an increase of Tregs differentiated from peripheral naive CD4+ T cells. Taken together this study also highlighted that the PD-1/PD-L1 system might contribute to the Treg/Th17 imbalance at the maternal-fetal interface [32].

By collecting peripheral blood, retroplacental blood, and cord blood after delivery in normal pregnancies, preeclampsia and gestational diabetes, Zhao et al. investigated the expression of different immune checkpoint molecules, e.g., PD-1, on T-cell subsets. Among other results they showed decreased PD-1 expression on T cell subsets in peripheral blood and retroplacental blood in preeclampsia and gestational diabetes compared to normal pregnancies. This may contribute to chronic inflammation seen in preeclampsia [33].

Another study by Meggyes et al. investigated the expression of different immune checkpoint molecules by MAIT (mucosal associated invariant T cell) and MAIT-like cells, an important proinflammatory and cytotoxic T-cell subset in peripheral blood and peripheral tissue-like liver or mucosa. They compared the blood of early onset preeclamptic patients to one of the healthy pregnant women. Besides a reduction of MAIT cells in preeclampsia, they found a higher activation and a significantly lower expression of PD-1 on these cells in the peripheral blood of preeclamptic patients. By this mechanism the MAIT cells escape from immune inhibition through binding of PD-L1, which might contribute to increased immunoactivation and thereby to the pathogenesis of preeclampsia [4].

Pianta et al. intended to investigate the role of human amniotic mesenchymal stroma cells (hAMSC), which were shown to have immunomodulatory and anti-inflammatory features, in the pathogenesis of preeclampsia. Among other results like inhibition of CD4/CD8 T-cell proliferation or induction of Treg cells, the study showed a higher expression of PD-L1 on the hAMSCs of preeclamptic patients. Taking the results together the authors concluded that the hAMSCs do not contribute to the pathogenesis of preeclampsia but quite the opposite might compensate the inflammation by their anti-inflammatory features [34].

The following Figure 1 summarizes the described alterations of the PD-1/PD-L1 system seen in preeclampsia and their effects on the immune system.

## 4. The TIM-3/Gal-9 System in Normal Pregnancy and Preeclampsia

### 4.1. The TIM-3/Gal-9 System in Normal Pregnancy

The T-cell immunoglobin and mucin domain-containing protein 3 (TIM-3) and its most relevant ligand Gal-9 are immune checkpoint proteins which are important for maintaining the equilibrium between proinflammatory and anti-inflammatory signals mediated by T cells. TIM-3 is a glycoprotein expressed on different immune cells like B cells, macrophages, monocytes, NK cells, T cells and mast cells as well as on antigen-presenting cells. Depending on the immune cell, binding of Gal-9 to TIM-3 can either lead to stimulatory or inhibitory effects on the immune system. Expressed on T cells the TIM-3/Gal-9 interaction leads to weakening of the Th1-mediated immunity and apoptosis of T cells which results in an inhibition of the immune system. On the other hand, when expressed on NK cells and dendritic cells the binding of Gal-9 results in an immunostimulatory effect [35].

Looking at the TIM-3/Gal-9 pathway in pregnant women, different studies showed an upregulation of the TIM-3 protein on leucocytes, especially on monocytes and NK cells, while the percentage on NKT-like cells and different T-cell subsets did not differ compared to nonpregnant women. Gal-9 levels on T cells were shown to be higher during pregnancy [9,36,37,38].

The TIM-3/Gal-9 pathway has been proven to promote maternal-fetal tolerance by inducing the development and expansion of regulatory T cells and Th2 cells and thereby leading to an anti-inflammatory effect of the immune system [39,40,41,42].

A blockade of TIM-3 results in an upregulation of pro-inflammatory cytokines as well as an accumulation of inflammatory granulocytes and macrophages at the maternal-fetal interface [41,43].

Sisti et al. investigated the status of the TIM-3/Gal-9 system in mononuclear cells of peripheral blood in twin pregnancies and compared it to the pH in the umbilical artery at delivery which thereby provides information about fetal outcome. The study showed elevated levels of both TIM-3 and Gal-9 in patients who suffered from fetal acidemia with a pH level < 7.15. In comparison the levels of both proteins were shown to be significantly lower when the pH level was ≥7.15. The authors concluded that elevations of these proteins in peripheral mononuclear cells could predict a low pH at delivery and thereby a worse fetal outcome [44].

### 4.2. The TIM-3/Gal-9 System in the Human Placenta

Several studies can be found about the involvement of the TIM-3/Gal-9 pathway at the maternal-fetal interface in the human placenta.

TIM-3 expression was shown to be evident in various lymphocyte subsets in the decidua throughout pregnancy. Looking at decidual NK cells, the majority showed high expression of TIM-3, leading to a mainly Th2 cytokine profile and thereby to an anti-inflammatory state of the immune system. Both, trophoblast and decidual tissue, can express large amounts of Gal-9. Blocking TIM-3 with a special fusion protein resulted in a reduced production of the pro-inflammatory cytokines IFN-γ and tumor necrosis factor (TNF)-α in decidual NK cells [9,45,46].

The expression of TIM-3 was not only shown on immune cells in the placenta but also on decidual stroma cells, leading to a higher production of Th2 cytokines and to an anti-apoptotic effect by preventing decidual stroma cells from apoptosis mediated by Toll-like receptors [9,47].

Hu et al. investigated the status of TIM-3 expression on Treg cells in mice, normal human pregnancy, and patients with recurrent pregnancy loss. They found a high number of TIM-3+ Treg cells in the decidua of mice as well as in normal pregnancy. TIM-3 expression on CD4+ T cells in the decidua was also shown to be evident. Compared to normal pregnancy the percentages of TIM-3+ CD4+ T cells, Treg cells, and TIM-3+ Treg cells in the decidua were reduced in patients with recurrent pregnancy loss. Looking at Gal-9 expression in placental villi, the study showed a significantly lower percentage in normal pregnancy compared to recurrent pregnancy loss. The secretion of IL-27 by trophoblast cells was significantly higher in normal pregnancy and resulted in the induction of TIM-3 expression on CD4+ T cells. The binding of Gal-9 then leads to the differentiation of these cells into TIM-3+ Treg cells and the promotion of apoptosis of T-effector cells in normal pregnancy [48].

Hutter et al. performed a study concerning the expression of tandem-repeat type galectins, including Gal-9, in human placenta collected in the third trimester in normal pregnancy and ones with intrauterine restriction of growth (IUGR). They showed an expression of the tandem-repeat type galectins in villous trophoblasts, extravillous trophoblasts, decidual stroma cells, and endothelial cells. Comparing the expression in normal pregnancy to pregnancy with IUGR, they found a significantly lower percentage of Gal-9 in the IUGR group carrying a male fetus. Pregnancies with a female fetus and IUGR were accompanied with an upregulation of Gal-9 [49].

The interaction of extravillous trophoblasts (EVTs) with decidual immune cells (DICs) is important for developing maternal-fetal tolerance and thereby maintaining pregnancy. Decidual immune cells can promote placental development and the function of extravillous trophoblasts. Li et al. demonstrated in their study that blockade of the TIM-3 and CTLA-4 pathways resulted in abnormal interaction of DICs-EVTs, thereby leading to poor placental development and increased fetal loss [50].

Taken together these studies show that the TIM-3/Gal-9 system is important for the establishment and the maintenance of maternal-fetal tolerance in early pregnancy.

### 4.3. The TIM-3/Gal-9 System in Preeclampsia

Whether the TIM-3/Gal-9 pathway as an important immune checkpoint system is relevant in the pathogenesis of preeclampsia is currently under research.

Miko et al. investigated the involvement of the TIM-3/Gal-9 pathway in the systemic inflammatory response which occurs in early onset preeclampsia. Therefore, they thoroughly studied the expression and function of both molecules on peripheral blood mononuclear cells. The expression of TIM-3 was shown to be significantly decreased by NK cells, CD56dim NK cells (=mature NK cells), T cells, and cytotoxic T cells in the preeclamptic group compared to healthy pregnant women. When looking at the Gal-9 expression they demonstrated increased Gal-9 levels in all investigated lymphocyte populations (T cells, cytotoxic T cells, NK cells, CD56dim NK cells, CD56bright NK cells, and NKT cells). The cytotoxic T cells and NK cells expressing TIM-3 in early onset preeclampsia showed increased cytotoxicity compared to the healthy pregnant group. Concluding these results, the authors suggest that the altered TIM-3/Gal-9 system may contribute to the enhanced systemic inflammatory response including the elevated pro-inflammatory Th1 response seen in preeclampsia [51].

The study of Hao et al. investigated the status of the TIM-3/Gal-9 system in decidual tissue at the RNA and protein levels in patients with preeclampsia. Using immunohistochemistry, they showed a primarily expression of both proteins on the cell membrane and in the cytoplasm, which were significantly higher in preeclampsia compared to normal pregnancy. The mRNA level of the proteins demonstrated an augmented but not significantly increased expression of TIM-3 and Gal-9 in the preeclamptic group. The expression of the pro-inflammatory cytokines IFN-γ and IL-17 was found to be significantly higher in the preeclamptic group, whereas the anti-inflammatory cytokine IL-10 was diminished. Analyzing the peripheral blood monocytes showed a higher TIM-3 expression in preeclamptic patients compared to normal pregnancies. Taken together these mechanisms may result in hyperimmune responses and thereby may play an important role in the pathogenesis of preeclampsia [52].

Wang et al. wanted to further analyze the involvement of this pathway in the onset of preeclampsia by comparing the expression of TIM-3 on different immune cells in the decidua of preeclamptic and healthy pregnancies and analyzing whether there is a change in TIM-3 expression in the peripheral blood of preeclamptic patients. They proved a significant downregulation of TIM-3 expression in the decidual immune cells in preeclampsia compared to normal pregnancy. The TIM-3+ decidual immune cells in healthy pregnancy showed higher production of anti-inflammatory cytokines and reduced production of pro-inflammatory cytokines such as IFN-γ and TNF-α compared to TIM-3 negative decidual immune cells. The blockade of TIM-3 leads to an enhanced production of the pro-inflammatory cytokine IFN-γ and a decreased production of the anti-inflammatory cytokines IL-4, IL-10, and GATA-3. It also resulted in an inhibition of tube formation. After administrating IL-4 and IL-10, the tube formation was promoted, which may prove that these reduced anti-inflammatory cytokines in preeclampsia could be responsible for the inadequate remodeling of spiral arteries. When looking at peripheral blood mononuclear cells of preeclamptic patients, a significant reduction of TIM-3 expression was shown on CD8+ T cells compared to normal pregnancies. This proves that TIM-3 is associated with immune tolerance in the third trimester of normal pregnancy and a disruption in the TIM-3/Gal-9 system may play a role in the pathogenesis of preeclampsia [53].

The study performed by Dong et al. investigated the involvement of TIM-3 and Gal-9 in the regulation of myeloid-derived suppressor cells (MDSC), which are myeloid progenitor cells able to suppress the activity of other immune cells, and its correlation with the pathogenesis of preeclampsia. The results showed a significantly higher expression of TIM-3 on monocytic MDSC in preeclamptic women compared to the healthy pregnant group. The monocytic MDSC produced significantly higher levels of the pro-inflammatory cytokine IFN-γ and significantly lower levels of the anti-inflammatory cytokine TGF-β in the preeclamptic group. Using immunohistochemistry, the expression of Gal-9 was found to be higher in the decidua and villi of preeclampsia-placentae compared to healthy pregnancies [54].

To analyze the role of TIM-3/Gal-9 in the polarization of decidual macrophages, Li et al. performed a study establishing a preeclampsia-like rat model by administering lipopolysaccharide. Besides an increase in pro-inflammatory and a decrease in anti-inflammatory cytokines, they observed increased M1 subtype and decreased M2 subtype in decidual macrophages at the maternal-fetal interface. The expression of TIM-3 and Gal-9 was shown to be reduced. The administration of recombinant Gal-9 leads to an attenuation of the PE-like manifestations in rats as well as to a shift of polarization of decidual macrophages toward an anti-inflammatory M2 subtype. The concentrations of TIM-3 and Gal-9 significantly increased after rGal-9 treatment, which highlights the importance of the pathway in early pregnancy and the involvement in the development of preeclampsia [55].

The following Figure 2 summarizes the described alterations of the TIM-3/Gal-9 system seen in preeclampsia and their effects on the immune system.

## 5. Conclusions

Preeclampsia is a pregnancy complication which affects 5 to 7% of all pregnant women and is accompanied with a high morbidity and mortality. Each year it is responsible for 70,000 maternal and 500,000 fetal deaths worldwide and is associated with an increased risk for cardiovascular and cerebrovascular diseases in women. The only effective treatment is the delivery of fetus and placenta [1]. Up until now the exact pathogenesis of the disease is still not fully understood, which is why further research is much needed. The immune checkpoint molecules are important key players in maintaining the balance of the immune system and thereby might also take part in the pathogenesis of preeclampsia. The summarized evidence in this review shows an existing alteration of the PD-1/PD-L1 system, both, in the periphery and at the maternal-fetal interface. Besides an increased cytotoxicity of T cells these alterations led to an imbalance of regulatory T cells and Th17 cells as well as an enhanced Th1 response. The soluble forms of both proteins were proven to be higher in preeclamptic patients compared to normal pregnancies, which might contribute to an augmented immune response. When looking at the maternal-fetal interface both proteins were shown to be downregulated in the placenta, leading to an increase in the Th17 production and thereby to a local imbalance of Treg/Th17 cells resulting in chronic inflammation. The TIM-3/Gal-9 system was shown to be modified in preeclampsia as well. In the periphery, a downregulation of TIM-3 and an upregulation of Gal-9 led to an increased cytotoxicity and an enhanced Th1 activation. TIM-3 was proven to be upregulated in MDSCs in the periphery, which resulted in an enhanced production of pro-inflammatory and a reduced production of anti-inflammatory cytokines. When looking at the maternal-fetal interface, studies showed an upregulation of TIM-3 and Gal-9 in the decidua leading to chronic inflammation by increasing the Th1- and Th17 immune responses as well as the production of pro-inflammatory cytokines. In contrary, TIM-3 was demonstrated to be downregulated in decidual immune cells, which also led to an increase in pro-inflammatory and decrease in anti-inflammatory cytokines. It also resulted in a shift of local macrophage polarization toward the pro-inflammatory M1-phenotype. Taken together, the listed alterations in the PD-1/PD-L1 system and the TIM-3/Gal-9 system in preeclampsia lead to a pro-inflammatory state of the immune system in the periphery as well as at the maternal-fetal interface and might thereby be involved in the pathogenesis of the disease. Further studies are needed to fully understand and specify the role of these immune checkpoint molecules in the pathogenesis of preeclampsia and to find a possible curative treatment of the disease.

## Figures and Tables

**Figure 1 medicina-58-00157-f001:**
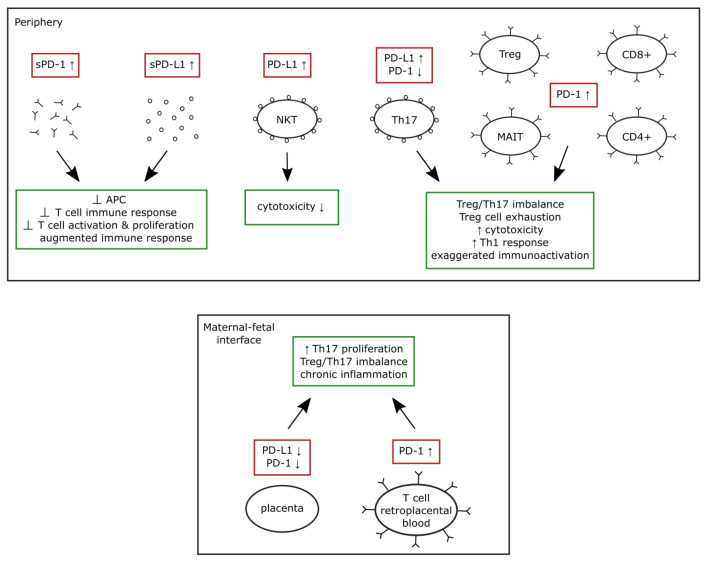
Summary of the described alterations of the PD-1/PD-L1 system in the periphery and maternal-fetal interface in preeclampsia and their effects on the immune system. Abbreviations: PD-1—programmed cell death protein 1; PD-L1—programmed death ligand 1; sPD-1—soluble programmed cell death protein 1; sPD-L1—soluble programmed death ligand 1; APC—antigen presenting cell; NKT—natural killer T cells; MAIT—mucosal associated invariant T cell; Th1—T helper 1 cell; Th17—T helper 17 cell; Treg—regulatory T cell; CD8+—CD8 positive T cell; CD4+—CD4 positive T cell; IL-17—Interleukin 17; IL-10—Interleukin 10; Ʇ—inhibition.

**Figure 2 medicina-58-00157-f002:**
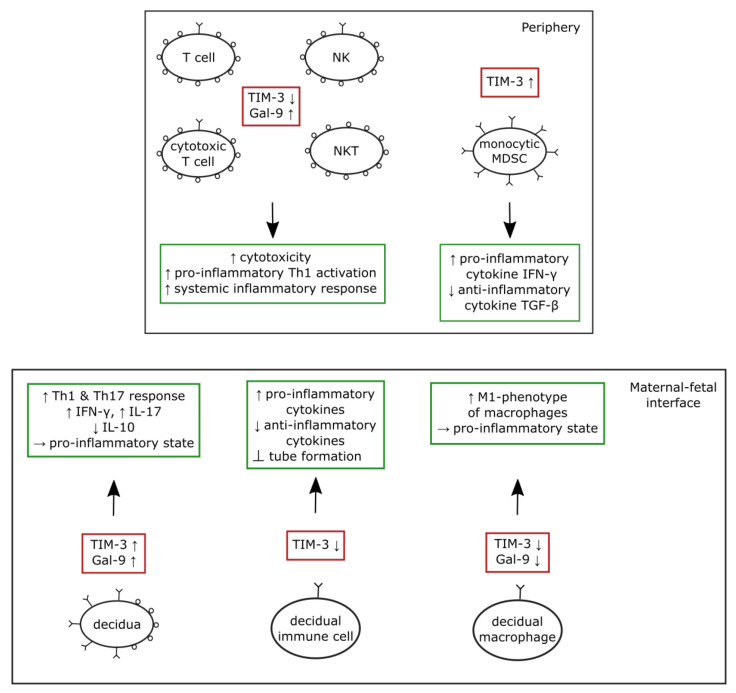
Summary of the described alterations of the TIM-3/Gal-9 system in the periphery and maternal-fetal interface in preeclampsia and their effects on the immune system. Abbreviations: TIM-3—T cell immunoglobin and mucin domain-containing protein 3; Gal-9—Galectin 9; NK—natural killer cell; NKT—natural killer T cells; MDSC—myeloid-derived suppressor cell; IFN-γ—Interferon-γ; TGF-β—transforming growth factor β; Th1—T helper 1 cell; Th17—T helper 17 cell; IL-17—Interleukin 17; IL-10—Interleukin 10; Ʇ—inhibition.

## Data Availability

All data were obtained from PubMed and they are freely available.

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
