# Peer review of "The Role of the Immune Checkpoint Molecules PD-1/PD-L1 and TIM-3/Gal-9 in the Pathogenesis of Preeclampsia—A Narrative Review"

_medicina, 2022, doi:10.3390/medicina58020157_

Round 1
Reviewer 1 Report
In the current review article entitled "The role of the immune checkpoint molecules PD-1/PD-L1 and TIM-3/Gal-9 in the pathogenesis of preeclampsia" authors have shown that alterations in the PD-1/PD-L1 system and the TIM-3/Gal-9 system in preeclampsia causes a pro-inflammatory state of the immune system in the periphery as well as at the maternal-fetal interface and might thereby be involved in the pathogenesis of the disease. The review is informative and incorporates relevant studies.
Author Response
Dear Reviewer 1,
thank you very much for your comments on my review “The role of the immune checkpoint molecules PD-1/PD-L1 and TIM-3/Gal-9 in the pathogenesis of preeclampsia”.
Concerning the English language and style: The review will be prove read by a native English-speaking college and corrected afterwards. The updated version will then be resubmitted.
Yours sincerely,
Johanna Mittelberger

Reviewer 2 Report
In my opinion, the Authors raise a important topic and relevant issue. I don't have any comments.
Author Response

(The authors gave the same response as above.)

Reviewer 3 Report
This review discusses the role of the immune checkpoint molecules PD-1/PD-L1 and TIM-3/Gal-9 in pregnancy and their alterations in preeclampsia. It is clearly organized and rich in content, and provides ideas for the future research of immunity immune checkpoint in pregnancy and pre-eclampsia.
Author Response
Dear Reviewer 3,
thank you very much for your comments on my review “The role of the immune checkpoint molecules PD-1/PD-L1 and TIM-3/Gal-9 in the pathogenesis of preeclampsia”.
Concerning the English language and style: The review will be prove read by a native English-speaking college and corrected afterwards. The updated version will then be resubmitted.
Yours sincerely,
Johanna Mittelberger
